# Characterising the Effect of Wnt/β-Catenin Signalling on Melanocyte Development and Patterning: Insights from Zebrafish (*Danio rerio*)

**DOI:** 10.3390/ijms241310692

**Published:** 2023-06-27

**Authors:** Praneeth Silva, Devi Atukorallaya

**Affiliations:** Department of Oral Biology, Dr. Gerald Niznick College of Dentistry, Rady Faculty of Health Sciences, University of Manitoba, Winnipeg, MB R3E 0W2, Canada; silvap1@myumanitoba.ca

**Keywords:** zebrafish, Wnt/β-catenin cell signalling, melanocytes, skin, pigmentation, neural crest cells derivatives

## Abstract

Zebrafish (*Danio rerio*) is a well-established model organism for studying melanocyte biology due to its remarkable similarity to humans. The Wnt signalling pathway is a conserved signal transduction pathway that plays a crucial role in embryonic development and regulates many aspects of the melanocyte lineage. Our study was designed to investigate the effect of Wnt signalling activity on zebrafish melanocyte development and patterning. Stereo-microscopic examinations were used to screen for changes in melanocyte count, specific phenotypic differences, and distribution in zebrafish, while microscopic software tools were used to analyse the differences in pigment dispersion of melanocytes exposed to LiCl (Wnt enhancer) and W-C59 (Wnt inhibitor). Samples exposed to W-C59 showed low melanocyte densities and defects in melanocyte phenotype and patterning, whereas LiCl exposure demonstrated a stimulatory effect on most aspects of melanocyte development. Our study demonstrates the crucial role of Wnt signalling in melanocyte lineage and emphasises the importance of a balanced Wnt signalling level for proper melanocyte development and patterning.

## 1. Introduction

The Wnt signalling pathway plays crucial roles in every aspect of embryonic development in vertebrates, including cell-fate specification, proliferation and differentiation, cell polarity, and morphogenesis [1,2]. The Wnt family, which includes 19 members in mammals and mice, is essential for embryonic development and tissue homeostasis [3], whereas the zebrafish genome contains 25 Wnt genes due to teleost-specific whole-genome duplication [4].

Wnt signalling plays a critical role in neural crest induction, neural crest cell (NCC) specification, NCC differentiation, and melanocyte development. Specifically, Wnt3a plays a crucial role in stimulating the generation of neural crest cells (NCCs) into melanocyte cells, and without this protein, NCCs are unable to differentiate into melanocytes [5,6]. Wnt3a signals cause an increase in melanocyte numbers by promoting the differentiation of melanoblasts into melanocytes, while Wnt3a and β-catenin are crucial for promoting the differentiation of NCCs into melanocytes [7,8]. Moreover, Wnt3a maintains and regulates the activity of MITF (microphthalmia-associated transcription factor) in melanoblasts, promoting the differentiation of melanoblasts into melanocytes [7].

Chromatophores, which are responsible for animal body coloration and are derived from NCCs, vary in number and type among different vertebrates [9,10]. While higher vertebrates such as mammals have only one type of pigment cell, called melanocytes [11], basal vertebrates such as zebrafish have several types, including iridophores, erythro-iridophores, xanthophores, leucophores, cyanophores, and erythrophores [9,10,12,13,14]. The evolution of body colouration and patterns serves various functions such as camouflage, sexual selection, and protection from UV radiation [15,16]. For ectothermic animals such as fish, amphibians, and reptiles, colour anomalies can affect animal fitness [17,18,19,20]. In zebrafish, melanocytes are found in both epidermal and dermal tissue, while in humans, epidermal melanocytes contribute to the pigmentation of hair and skin, but not to rapid colour changes [21].

Zebrafish have two methods of melanocyte development during the embryonic stage, directly from NCCs and from melanocyte stem cells [22]. Early larval stage melanocytes come from both the remaining embryonic melanocytes and NCCs [23]. Zebrafish embryos develop their first melanocyte at approximately 30 h post fertilisation (hpf) [24]. Zebrafish epidermal melanocytes are round, while dermal cells are dendritic when immature [21,25]. By 2 days post fertilisation (dpf), melanocytes mature and flatten out, playing a role in establishing mid-larval, juvenile, and adult pigment patterns [23]. Melanocyte stem cells migrate along the dorso-lateral and ventro-medial pathways to generate embryonic and post-embryonic melanocyte strip formation [26]. By 6 dpf, the early larval pigment pattern is established and completed, typically consisting of four longitudinal stripes. The proper arrangement of melanocytes along the left–right (L–R) axis of the fish body is crucial for the formation of a perfect pigment pattern [27,28].

The process of melanocyte development is similar in all vertebrates [29]. However, the appearance and arrangement of melanocytes can be influenced by external factors [11]. Melanocytes can undergo changes in their development in response to various environmental stimuli such as exposure to various drugs, UV radiation, changes in hormones, and substances that affect cell signalling pathways involved in melanocyte development [30].

Since the Wnt signalling pathway plays an important role in regulating the main aspects of cell development, abnormal Wnt signalling can be detrimental and obstructs the proper development and function of body organs and systems in any organism [31]. Therefore, Wnt signalling inhibitors and activators are administered to identify mechanisms of the Wnt signalling pathway in health and disease progression [32].

Zebrafish is an important model organism in melanocyte research because it shares many similarities with humans in terms of the genes and molecular mechanisms involved in melanocyte development and function [29]. Additionally, zebrafish embryos are transparent, which makes it easier to study the formation and migration of melanocytes in vivo (Figure 1A). The melanocyte development in zebrafish is also very rapid, and the embryos can develop pigmentation within a few days after fertilisation (Figure 1B) [28,33].

Research on melanocyte development has previously focused on studying molecular pathways and the effects of drug exposure in various animal models, such as zebrafish, avian embryos, murine models, and *Xenopus laevis* embryos [7,34,35,36,37]. However, to date, there has been no investigation into the effects of early embryonic exposure to Wnt signalling chemical modulators on the fundamental aspects of melanocyte biology during development and maturation. While previous studies have shed light on the molecular mechanisms underlying melanocyte development and the impact of drug exposure on this process, the effects of early embryonic exposure to Wnt signalling chemical modulators remain unknown. This is a significant gap in our understanding of melanocyte biology, as early embryonic exposure to these chemicals could potentially alter the fundamental aspects of melanocyte development and function, leading to long-term consequences for skin pigmentation and melanoma risk.

Therefore, investigating the effects of early embryonic exposure to Wnt signalling chemical modulators on melanocyte development is crucial. Understanding how these chemicals affect melanocyte biology over developmental stages and until maturation will provide insight into the potential risks associated with exposure to these chemicals during pregnancy and early development.

Wnt signaling plays a critical role in neural crest induction, NCCs specification, NCCs differentiation, and melanocyte development. In particular, Wnt3a plays a critical role in stimulating the generation of NCCs into melanocyte cells, and without this protein, NCCs are unable to differentiate into melanocytes [5,6]. Wnt3a signals cause an increase in the melanocyte numbers by promoting melanoblasts to melanocytes, while Wnt3a and β-catenin are crucial for raising the differentiation of NCCs into melanocytes [7,8]. Moreover, *Wnt3a* maintains and regulates the MITF activity of melanoblasts in order to promote melanoblast differentiation into melanocytes [7].

LiCl (Lithium Chloride) is one of the most prominent Wnt signalling activators that work on inhibiting the GSK-3β enzyme, which normally interrupts the formation of the β-catenin destruction complex, permitting translocation of β-catenin into the nucleus to participate in gene transcription and expression (Figure 2) [38]. Wnt-C59 is a strong PORCN inhibitor. This protein includes the membrane-bound O-acyltransferase (MBOAT) family. However, the inhibition of the Wnt signalling pathway is caused by the obstruction of the palmitoylation mechanism of the Wnt protein, conducted by the PORCN enzyme (Figure 2) [39].

This study aimed to investigate the impact of early embryonic exposure to Wnt chemical modulators on melanocyte development using zebrafish as an experimental model. The hypothesis was that early embryonic exposure to small molecules of Wnt signalling regulators (agonists and antagonists) affects melanocyte formation, phenotype, migration, and arrangement. The differences in melanocyte count, phenotype, migration, four-strip arrangement, and L–R biased migration in each chemical-exposed group were examined after administering Wnt chemical modulators to the embryos. The results of the study indicate that inhibition of the Wnt pathway during early embryonic stages negatively affects skin melanocyte formation and patterning, while stimulation of the Wnt pathway promotes key aspects of melanocyte development in zebrafish.

## 2. Results

### 2.1. Microscopic Examination of Melanocyte Formation of the Control and Wnt Chemical-Treated Zebrafish Larvae

The effect of Wnt signalling alterations on melanocyte development in zebrafish was studied using stereo-microscopic examinations. All the chemical-exposed fish (Figure 3E–L) showed changes in melanocyte formation at the head region compared to the control sample. Considering the melanocyte counts in the head region of the LiCl (Figure 3E–H) exposed embryos, higher values were observed within the ROI of the selected 4, 6, 8, and 10 dpf developmental stages compared to the untreated control samples. Conversely, lower melanocyte formation was observed in the W-C59 (Figure 3I–L) exposed fish group.

### 2.2. Comparison of the Variation in Melanocyte Count of the Control and Wnt Chemical-Treated Zebrafish Larvae

Based on the analysis of the phenotypic data, we observed higher values for melanocyte count in LiCl-treated zebrafish larvae at each developmental age compared to the control sample (Figure 4A). Interestingly, the fluctuation of melanocyte densities in the LiCl-exposed fish was not like that of the control sample. In the melanocyte count analysis, we observed sudden fluctuations in melanocyte numbers in W-C59 exposed fish groups during every life stage, in contrast to un-manipulated fish (Figure 4B).

### 2.3. Comparison of the Variation of Melanocyte Count at 8 dpf against Wnt Chemical Treatments

Melanocyte count was investigated at the 8 dpf stage of zebrafish, which is a crucial stage in the melanocyte development lineage of the fish between the 4 dpf to 10 dpf growth period. Zebrafish larvae exposed to LiCl (40 ± 0.86) at the embryonic stage showed a higher melanocyte count than the control fish, while a lower melanocyte count was observed in W-C59 (28.2 ± 2.88) exposed fish. However, there was no significant difference (*p* > 0.05) in mean melanocyte counts among the control and chemical-treated larvae (Figure 5).

### 2.4. Comparison of the Variation of Melanocyte Count at 15 dpf against Wnt Chemical Treatments

The effect of Wnt chemical exposure on melanocyte development at the mid-larval stage (15 dpf) was investigated. Notably, high melanocyte count was observed in the LiCl-treated (66.5 ± 1.64, *p* > 0.05) fish, while lower melanocyte count was observed in W-C59 exposed (57.9 ± 2.12) fish. Significant differences in the mean melanocyte counts were observed between W-C59 exposed fish and the control group (*p* ˂ 0.01) (Figure 6).

### 2.5. Comparison of the Differences in Melanocyte Morphology and Morphometry against Wnt Chemical Treatments

#### 2.5.1. Melanocyte Phenotypic Differences at 2 dpf Embryonic Stage of Zebrafish

Contrast changes in melanocyte morphology were detected in zebrafish embryos exposed to chemical treatments, compared to the control sample. In the control sample, melanocytes were thin plaque-shaped cells with defined margins (marked by black arrowheads). However, in embryos treated with W-C59, most melanocytes showed a lack of pigmentation and appeared pale (Figure 7). On the other hand, the pigment intensity and morphology of the melanocytes in embryos treated with LiCl were like those in the head region of the control sample (Figure 7).

##### Differences in Melanin Dispersion at 2 dpf Embryonic Stage of Zebrafish

Differences in melanin dispersion were observed in the control and chemical-exposed embryos, as illustrated in the bar chart below. Melanin dispersion was measured using microscopic software analysis as an indicator to determine melanosome dispersion. A significant difference was recorded in the values of mean melanin dispersion between control and different chemical exposures of fish (*p* ˂ 0.001). According to the data analysis, less melanin dispersion was found in the W-C59-treated (796.84 ± 28.43) fish compared to normal rearing control zebrafish. In contrast, high melanin dispersion was recorded in the LiCl-exposed fish (924.78 ± 68.74) over W-C59-treated fish (Figure 8).

#### 2.5.2. Melanocyte Phenotypic Differences at 6 dpf Larval Stage of Zebrafish

Microscopic examinations of the chemical-treated larval stage zebrafish showed numerous deficits in melanocyte morphology. Fish exposed to W-C59 (marked by dark blue arrowhead of Figure 9) showed deviations from the typical oval/round shape of melanocytes seen in the control fish. Instead, polygonal shapes were observed. However, no significant differences in paleness of melanocytes were noted at the 6 dpf stage, except for the observation of small pores in LiCl-treated fish. In general, the melanin synthesis and pigment distribution in LiCl-treated fish (dark blue arrowhead) were like those of the control group (Figure 9).

##### Differences in Melanin Dispersion at 6 dpf Larval Stage of Zebrafish

The analysis of the physical characteristics of larval fish at 6 dpf showed that exposure to Wnt regulatory chemicals during embryonic development resulted in abnormalities in melanin dispersion. The mean area of melanin dispersion was significantly different in fish that were exposed to different chemicals, with LiCl-treated fish (1757.53 ± 36.52) having higher melanin dispersion compared to W-C59-treated fish (1311.04 ± 50.78). These differences were statistically significant, with a *p*-value of less than 0.001 (Figure 10).

### 2.6. Examination of Melanocyte Migratory Differences against Wnt Chemical Treatments

Characteristic defects were observed in both migratory and non-migratory sites of the chemical-treated embryos (Figure 11) (refer to Figure 17. In the LiCl-treated fish, the highest melanocyte accumulations were recognised in both migrated sites: the yolk sac region (marked by black arrowhead) and yolk sac extension (dark blue arrowhead), as well as some non-migrated sites (Figure 11). The presence of high melanocyte count in the LiCl-treated embryos was greatly evident by their appearance, which was somewhat darker in colour due to dispersed melanocyte arrangement throughout the embryos. Wnt inhibitor-treated (W-C59) fish produced fewer migrated melanocytes in the regions of the anterior head and yolk sac regions. At the same time, the least melanocyte count was recorded near the otic vesicle (Figure 11).

### 2.7. Analysis of Melanocyte Migratory Differences against Wnt Chemical Treatments

There were significant differences in both mean melanocyte densities of migrated and non-migrated fish that were exposed to chemicals (*p* < 0.001) (Figure 12). The highest non-migrated melanocyte count (19.1 ± 12.04, *p* < 0.001) and migrated melanocyte count (61.3 ± 4.54, *p* < 0.001) were observed in LiCl-treated embryos. W-C59 treatment resulted in lower migrated melanocyte counts (36.1 ± 2.43, *p* > 0.05) and higher non-migrated melanocyte counts (10.6 ± 1.40, *p* < 0.001) than the control. Interestingly, all values for non-migrated melanocyte counts in chemical-treated embryos were significantly different from the control sample (*p* < 0.001), while LiCl-treated embryos were only significant for migrated melanocytes (Figure 12).

### 2.8. Variations of Melanocyte Arrangement in Stripe Formation against Wnt Chemical Treatments

#### 2.8.1. Variations of Dorsal Stripe Melanocyte Arrangement

Microscopic images of the dorsal view of the fish revealed changes in melanocyte arrangement during dorsal stripe formation in response to chemical exposures (Figure 13). Compared to the control sample, lower melanocyte counts were observed within the stripes of LiCl and W-C59 chemical-exposed fish (marked by dark blue arrowheads). A relatively reduced melanocyte density was observed in the dorsal stripes of W-C59-treated larval fish compared to LiCl-treated fish, with aberrations in melanocyte shape and discontinuous melanocyte arrangement.

#### 2.8.2. Variations of Yolk Sac Stripe Melanocyte Arrangement

At the early larval stage of zebrafish, the yolk sac stripe had a distinct diamond-shaped arrangement in the control sample (indicated by the dark blue arrowhead in Figure 14). The typical shape of the yolk sac stripe was also altered in the treated samples, while the stripe phenotypes of the LiCl and W-C59-treated fish were fairly like the control stripe formation (Figure 14). Melanocytes in the yolk sac stripe of the LiCl-treated fish were more dispersed (dark blue arrowhead), while the W-C59-treated fish had a considerably aggregated phenotype compared to the uniform arrangement of the control sample (Figure 14).

#### 2.8.3. Analysis of Melanocyte Arrangement at Early Larval Stage upon Wnt Chemical Treatments

The contribution of melanocytes to each dorsal, lateral, ventral, and yolk sac stripe formation was quantified by counting, as shown in the bar chart presented in Figure 15. The melanocyte counts in each stripe formation of chemical-treated fish were comparatively reduced, apart from the yolk sac stripe, which showed an increased melanocyte count in the LiCl-treated fish relative to the control sample. However, exposure to small molecules of Wnt chemical modulators had a significant effect on melanocyte stripe formation compared to the control sample (*p* < 0.001) (Figure 15).

### 2.9. Analysis of Biased Migration of Melanocytes along the L–R Axis of Zebrafish Larvae upon Wnt Chemical Treatments

The biased migration of melanocytes along the L–R axis in untreated control and chemical-treated fish groups was analysed by focusing on the melanocyte counts along the lateral stripe of the fish. No changes in the L–R symmetry of melanocyte arrangements were observed in the untreated control larvae. However, upon examining the captured images of the L–R axes of the LiCl- and W-C59-treated larvae, as shown in Figure 16, increased melanocyte counts prominently displayed on the left side axis compared to the right axis of the fish body were found. Interestingly, all the fish examined in the LiCl treatment showed a trend toward left-sided melanocyte migration, while the W-C59 chemical-treated fish showed biased melanocyte migration on both L–R axes. However, exposure to LiCl had a significant effect on biased melanocyte migration along both L–R body axes of fish (*p* < 0.001), but this effect was not detected in W-C59-exposed fish (Figure 16).

## 3. Discussion

The effect of early embryonic exposure to small molecules of Wnt regulatory chemicals on melanocyte formation, phenotypic development, migration, and patterning has not been observed in any studies using any animal model, including zebrafish [40,41,42]. Exclusively, this is the first study to investigate the effect of early embryonic Wnt signalling activity on zebrafish melanocyte development and patterning.

In the present study, LiCl exposure resulted in high melanocyte counts in the dorsal head region of treated fish (Figure 3 and Figure 4) compared to melanocyte development in the control group. LiCl is widely used in research as an agonist of the canonical Wnt signalling pathway, mainly by stabilising free β-catenin in the cytosol by inhibiting the GSK-3β activity, which ultimately stimulates the canonical Wnt/β catenin signalling pathway [38,43]. Research on both mice and zebrafish has revealed that the expression of MITF (MITFa in zebrafish) in NCCs is vital for melanocyte specification [44,45]. As mentioned earlier, MITF/MITFa regulates all the key aspects of melanocyte cell biology, including specification, proliferation, and differentiation of melanoblasts as well as development, morphology and melanogenesis and the survival of melanocytes themselves [46,47]. However, MITF transcription is mainly controlled by the Wnt signalling, through the LEF and β-catenin-mediated regulation [48,49]. Studies that involved mammalian melanocytes have shown that the inhibition of GSK-3β activity results in the upregulation of MITF expression and differentiation in normal human melanocytes [50]. Bromoindirubin-3′-oxime (subsequently referred to as BIO) is a well-characterised GSK-3β inhibitor and has been used to investigate the effects of increased Wnt signalling in many model systems [51,52]. Melanocyte development of zebrafish has been examined using the same Wnt activator by exposing it to zebrafish embryos and revealed that it increases the melanocyte counts in the head and trunk regions of the 3 dpf fish [53]. The results of the current study reflect the effect of LiCl on stimulating Wnt signalling and the developmental processes of early zebrafish neurulation, specifically on the specification of NCCs in melanoblast development [5]. Further, early experiments demonstrated that the overexpression of β-catenin in NCCs has raised the melanocyte cell generation instead of other neuronal cells [5]. This present study adds further support to the previous research findings mentioned above.

W-C59 is considered one of the most potent inhibitors of the Wnt signalling pathway and has been applied to various model systems; particularly attenuating the signalling cascades of fibrotic disorders and reducing the effects of kidney fibrosis [39,54]. This drug prevents the Wnt target genes which are involved in diseases via interrupting β-catenin signaling. Further, this chemical has been used in mouse models to suppress the growth of nasopharyngeal tumours and arrest cancer stem cells [55]. Evidence has been found that increased Wnt/β-catenin signalling is associated with tumorigenic pathways in breast cancer cells and tumor progression can be suppressed with the application of Wnt- antagonist, W-C59 as a therapeutic approach to patients [56].

In the current study, this pharmacological agent was applied for the first time in zebrafish research to investigate the effect of Wnt signalling inhibition on melanocyte development (Figure 3 and Figure 4). Early embryonic exposure to W-C59 displayed sudden ups and downs in melanocyte counts in the head region over the 4–10 dpf fish development period (Figure 4). This phenomenon could be due to the direct effect of W-C59 on melanocyte specification, differentiation, and Wnt-related genes. To our knowledge, this study is the first to demonstrate that the early embryonic exposure of zebrafish (*Danio rerio*) to Wnt chemical modulators, including LiCl and W-C59, during the early neurulation period, has an impact on the generation of melanocytes. These findings highlight the importance of Wnt signalling in the development of melanocytes in zebrafish.

The 8th day of zebrafish development is considered a crucial stage in postembryonic growth, marked by several changes in the pigment pattern [22,57]. Zebrafish display new features in pigment pattern development, remodelling and losing some chromatophores that arose at earlier stages to generate stripes and interstripes [58]. New melanocytes and iridophores also enter into the pattern formation while existing in the early larval melanocytes [58].

Differences in melanocyte formation shown in Figure 5 indirectly reflect the effect of embryonic chemical exposures on the formation of new melanocytes that arrive at the larval stage of fish. The results of this study provide evidence of the stimulatory effect of LiCl on the development and survival of melanocyte stem cells, as well as the differentiation of melanocytes (Figure 5). In contrast, Wnt signalling inhibition (W-C59) has a negative effect on melanocyte cell differentiation compared to the control and LiCl chemical exposure (Figure 5).

New melanocytes start to emerge around the 14th day of fish development, which is when metamorphosis begins. At this stage, the number of melanocytes in fish gradually increases, and they start to spread evenly throughout the entire body [59]. The significantly lower melanocyte counts in the W-C59-treated fish (Figure 6) indicate a persistent inhibitory effect on melanoblast generation and/or melanocyte differentiation even in mid-larval stages. The LiCl-treated fish group also displayed a non-significantly higher melanocyte count (Figure 6) compared to the control group, suggesting the possibility of a stimulatory effect on melanocyte development persisting until post-embryonic stages. Overall, the results of this study indicate that Wnt signalling stimulation or inhibition at the early neurulation stage of zebrafish embryos has a prolonged effect on melanocyte formation rather than a short one.

During the early embryonic stage of zebrafish, immature melanocytes have a highly dendritic cell morphology [11], which indicates that they are in the initial stages of melanin production. On the other hand, mature melanocytes have a flattened and thin plaque-like morphology [11], as shown in Figure 7. Various environmental factors can cause changes in this melanocyte phenotype [9,60]. In the current study, less melanin pigmentation and paleness were displayed in the melanocytes of the W-C59-exposed (Figure 7) fish group. Defects in the melanocyte phenotype could result from improper differentiation of melanocytes, including the inhibition of Wnt signalling and melanogenic gene expression (TYR, TYRP1, and Dct) [11]. However, displaying a similar phenotype of LiCl-incubated fish with the control group (Figure 7), indicates that LiCl supports or has a less negative impact on melanocyte differentiation.

All the fish exposed to LiCl and W-C59 showed a significant decrease in melanosome dispersion compared to untreated fish, as shown in Figure 8. The melanin dispersion of melanocytes is determined by the aggregation and dispersion of melanosomes along melanosome motor activity, unlike melanin formation [61]. The analysis of melanin dispersion at the embryonic stage indicates that both Wnt cell signalling activation and inhibition resulted in aggregating melanosomes toward the centre of the cells, rather than dispersing toward the periphery [62,63,64]. This effect was particularly noticeable in the melanocytes of fish exposed to Wnt inhibitors, as depicted in Figure 8.

To investigate the effects of Wnt signalling activation and inhibition on melanocyte development at the early larval stage, fish at 6 dpf were examined. The LiCl- and W-C59-treated fish showed defects in melanocyte morphology and arrangement in the head region (Figure 9), indicating that these chemicals can affect melanocytes even at later developmental stages. The changes in the melanocyte phenotype were more pronounced in the W-C59-exposed fish group, except for the LiCl-treated fish whose melanocytes were like those of the control group (Figure 9). However, the melanin dispersion was reduced in both treated groups compared to the control group (Figure 10), suggesting that these chemicals can modulate the regulatory mechanism of intracellular melanosome dispersion and alter the fish’s external coloration. However, the defects in embryonic and larval melanocyte morphologies may also result from direct chemical effects on the central nervous system and melanocyte stem cells.

During embryonic development, melanocytes migrate along two different migratory routes. Melanocytes that move along the dorsolateral pathway are responsible for the melanocytes found in the embryonic lateral stripe, head, yolk sac, and yolk sac extension regions. On the other hand, the ventromedial pathway gives rise to melanocytes in the head, yolk sac, and yolk sac extension regions, with the exception of the lateral stripe [23,65]. However, changes in both extrinsic and intrinsic factors that are important for melanocyte migration could lead to defects in melanocyte migration. This issue was investigated in the present study, which revealed that embryonic exposure to Wnt signalling modulators caused migratory defects during the embryonic stage (Figure 11 and Figure 12). Data analysis showed a significant increase in the counts of migrated and non-migrated melanocytes after LiCl exposure (Figure 12), with supernumerary melanocytes being observed in the head, yolk sac, and yolk sac extension regions of the embryonic fish (Figure 11). The high melanocyte counts observed after LiCl exposure might be due to elevated levels of Wnt signalling during the early neurulation period [66,67]. The inhibition of Wnt signalling during the embryonic stage resulted in a reduction in the migrated melanocyte count and a significant increase in the non-migrated melanocyte number (Figure 12). This finding suggests that a low level of Wnt signalling does not support melanocyte migration [68]. Past studies have recognised the importance of maintaining Wnt expression from the onset of neural crest emigration until completion for proper melanocyte migration in vertebrate embryos [8]. Thus, Wnt signalling expression at the right time and at accurate levels affects the proper melanocyte migration of animals [69].

The patterns of embryonic melanocytes in zebrafish, established by 2 dpf, play a crucial role in directing the development of the early larval pigment pattern at 6 dpf [23]. Any aberrations in melanocyte stripe formation during this process are mainly due to defects resulting from embryonic melanocyte migration. Specifically, melanoblasts that follow the dorsolateral pathway contribute to the formation of all four larval stripes, while those that take the ventromedial route migrate through the horizontal myoseptum and form all the larval stripes except for the lateral one [23,65]. As a result, differences in melanocyte patterns in dorsal and yolk sac stripe formation are a result of errors in the migratory pathways of melanocytes (Figure 13 and Figure 14). Reduced melanocyte counts were observed in the four-stripe formation of W-C59-treated larval fish, while comparably high melanocyte counts were noticed in LiCl-exposed fish. The lower melanocyte development in the W-C59-treated fish group (Figure 13, Figure 14 and Figure 15) indicated that Wnt inhibition severely affects melanocyte migration along both migratory pathways. High counts of migrated melanocytes were previously recorded in the yolk sac and yolk sac extension regions of LiCl-exposed fish (Figure 11), which could direct yolk sac stripe formation in 6 dpf larval fish with high melanocyte numbers (Figure 14 and Figure 15). Overall, the present study presents an exciting opportunity for future research to investigate how early embryonic exposure to Wnt signalling modulators affects melanocyte pattern formation in zebrafish.

Exposure of Wnt cell signalling chemical modulators to zebrafish embryos demonstrated biased melanocyte localisation along the L–R axis of the larval fish body (Figure 16). Most of the previous studies on L–R asymmetry have been focused on the mechanisms underlying that determine the laterality of major body organs: including cardiac, visceral, and other mesodermal derivatives using zebrafish and other numerous vertebrate models [70,71,72].

Organ lateralisation of vertebrates is determined by changes in several signalling cascades, including Wnt, Nodal-Pitx2 [73], and Lefty1/Lefty2 [61,62,63,64]. Kupffer’s vesicle (KV) is the organiser region responsible for establishing left-right (L–R) asymmetric patterning in zebrafish. These signalling pathways are also involved in key steps of organogenesis and L–R patterning organisers of teleosts [74]. However, no studies have focused on examining the L–R patterning of neural crest-derived organs. In our study, we observed the localisation of different numbers of melanocytes at the lateral L–R trunks of LiCl and W-C59-treated fish (Figure 16). According to the literature, the development of different melanocyte numbers at lateral L–R trunks of the larval fish observed in our study could be a result of L–R biased differentiation of NCCs into melanocytes or due to biased migration of differentiated melanocytes with embryonic chemical exposure [75]. It could be possible to take place biased melanocyte migration as recorded different melanocyte numbers on lateral stripe with differential chemical exposures. The results of the present study declare the information that L–R asymmetry is not unique to the visceral and neurally-derived tissues but also extends to the behaviour of migratory neural crest derivatives.

A similar asymmetry was observed in the neural crest-derived melanocytes of *Xenopus* embryos [75]. The lateral organization of melanocytes occurs by following the dorso-lateral migration paths along the sides of the zebrafish embryos while differentiating from NCCs. Inasmuch, the bias melanocyte migration can arise from the defects in L–R lateral migratory routes. Interestingly, the same phenomenon has been examined in the previous study, which revealed that there is no apparent asymmetry in the extent of melanocyte differentiation from the NCCs prior to the onset of migration; by contrast, L–R asymmetry in the number of melanocytes was found after the migration [75]. This further suggests that the observed asymmetry is due to the asymmetric migration patterns.

Remarkably, we recorded an 80% left-sided biased migration in LiCl-incubated fish, but mixed L–R biased migration was observed with W-C59 exposure (Figure 16). These data indicate the significant influence of Wnt signalling regulatory chemicals on the symmetric migration of neural crest derivatives. Future studies are needed to experiment with the mechanisms that govern the L–R asymmetrically biased response of cells upon the effect of embryonic chemical exposures.

## 4. Materials and Methods

### 4.1. Zebrafish Strains and Maintenance

These experiments were conducted on male and female wild-type AB zebrafish (*Danio rerio*) purchased from Zebrafish Genetics and Disease Models Core Facility, Hospital for Sick Children (Toronto, ON, Canada). The fish were housed, and their colony was established according to Institutional Animal Care and Use Committee protocols at the central animal care facility at the Bannatyne campus, University of Manitoba. Adult zebrafish were fed a diet of Gemma 300–supplemented live shrimp and maintained on a 14/10 day/night cycle. Embryos were obtained from natural spawning. The collected eggs were cleaned and reared according to standard conditions. All the experimental protocols and guidelines for setting up fish breeding, collecting, and cleaning eggs and rearing embryos in this research were followed according to the Canadian Council of Animal Care (CCAC) and under the protocol number 17-041 (AC11315) by the Animal Care Committee, University of Manitoba.

### 4.2. Chemical Treatment of Wnt Modulators

Zebrafish embryos were treated with Wnt pathway agonist and antagonist chemicals at 10 hpf in glass Petri dishes. The chemicals used were 2 mM LiCl (Cat. No. 866405-64-3; TCI America, Portland, OR, USA) as the Wnt pathway agonist and 10 nM W-C59 (Cat. No. 500496; Sigma-Aldrich, Oakville, ON, Canada) as the Wnt pathway antagonist. Control samples were kept in fish water. The embryos were incubated in the chemical treatments for 12 h and washed 3–4 times with fish water at 22 hpf. The embryos were raised at 28.5 °C, and the water was changed once every day. They were euthanised at different ages using 1% tricaine methanesulphonate (MS222) (Cat. No. 118000500; Acros Organics, Morris Plains, NJ, USA) and fixed overnight in 4% paraformaldehyde (PFA), then stored in phosphate-buffered saline (PBS). For each experiment, 10 fish were used.

### 4.3. Analysing the Melanocyte Count

Zebrafish were analysed between 4 and 10 dpf to investigate the effect of alterations in Wnt signalling on melanocyte count. This methodology was adopted from our previous publication [38].

Briefly, embryos/larvae were mounted in 0.2% agar in embryo medium without a coverslip, and a region of interest (ROI) was imaged in fixed fish using a Zeiss Discovery V8 stereomicroscope. Melanocyte count was analysed within the ROI, which encompassed the dorsal view of the head from midway up the eyes to the base of the head. Cells with an area of ≥50% within the ROI were included, and cells were counted using the ZEN 2011 software 3.6 (blue edition, Zeiss, Oberkochen, Germany, 2011).

### 4.4. Examining Morphology and Measuring Melanin Dispersion of Melanocytes

The morphology (shape and melanin formation) and area of melanin dispersion of melanocytes were investigated in control, Wnt activator, and Wnt inhibitor-treated zebrafish groups at the embryonic (2 dpf) and early larval (6 dpf) developmental stages. The area of melanin dispersion of melanocytes within the region of interest (ROI) was measured along the perimeter of melanocytes using ZEN 2011 software tools (version 3.6). The mounting and imaging procedures were conducted as described previously.

### 4.5. Analysing the Melanocyte Migration

Migrated and non-migrated melanocytes were determined by counting the melanocytes at 48 hpf. Melanocyte migration was analysed in each control, Wnt activator, and Wnt inhibitor-treated group along six distinct regions of the embryo: the anterior head, yolk sac region, yolk sac extension, near the ear, the region between the dorsal stripe and horizontal myoseptum, and the region between the ventral stripe and horizontal myoseptum Figure 17. Melanocytes located on the anterior head (A), yolk sac region (B), and yolk sac extension (C) were classified as a migrated subpopulation of melanocytes and are coloured in yellow in Figure 17 Melanocytes that remained near the ear (D), the region between the dorsal stripe and horizontal myoseptum (E), and the region between the ventral stripe and horizontal myoseptum (F) were marked as orange in Figure 17 and were referred to as a non-migratory subpopulation of melanocytes. Statistical analysis was conducted to check whether Wnt signalling modulation had a significant effect on the number of migrated and non-migrated melanocyte cells. Here, the ratio of migrated to non-migrated melanocyte subpopulations in Wnt activator and Wnt inhibitor-treated fish embryos were compared to that of the control.

### 4.6. Analysing the Melanocyte Arrangement

Melanocyte arrangement and quantification of melanocytes in each stripe were carried out at the 6 dpf early developmental larval stage when the four-striped melanocyte pattern is established (Figure 1B), based on microscopic examinations. At this stage, the melanocyte pattern consists of four stripes of melanocytes (lateral view): the dorsal stripe, the lateral stripe, the ventral stripe, and the yolk sac stripe, parallel to the anteroposterior axis along the horizontal myoseptum. The Wnt chemical exposure on the later development of four-stripe melanocyte arrangement was analysed by comparing it with the control. Differences in melanocyte pattern and count in each stripe due to Wnt signalling activation and inhibition were captured using the Zeiss discovery V8 stereomicroscope.

#### Analysis of Biased Migration of Melanocytes along the L–R Axis

Biased migration of melanocytes was determined along the left–right (L–R) axis of the lateral melanocyte stripe at 6 dpf larval fish (Figure 1B). The melanocyte cells on both the left and right sides of each larval fish were counted, and the direction of bias (either leftward or rightward) was quantified in each group of control, Wnt activator, and Wnt inhibitor-treated fish.

### 4.7. Statistical Analysis

Statistical analyses were performed using Excel (2016, Microsoft, Redmond, Washington, USA). Graphical representations were also designed with the same software. A one-way ANOVA test was carried out to determine the significant difference in means between the independent groups of the associated variable.

Significant main effects were further decomposed using pairwise comparisons with a post hoc Simple Bonferroni correction with Student’s *t*-test, for multiple comparisons. Data were expressed as Mean ± SEM, and a probability level of 5% was used as the minimal criterion of significance. *p* > 0.05 was considered non-significant while the *p* values of ≤0.001 (***), ≤0.01 (**), and ≤0.05 (*) were taken as statistically significant. * Depicts the magnitude of the significance.

## 5. Conclusions

This research identified that altered Wnt signalling activity during the embryonic early neurulation period influences melanocyte formation, differentiation, and defects in melanocyte migration and arrangement. Zebrafish provide a useful model for investigating the effects of the Wnt signalling pathway on the development and patterning of neural crest cell-derived melanocytes. Wnt inhibition during the early neurulation period leads to the downregulation of the melanocyte lineage, while Wnt signal stimulation promotes most aspects of melanocyte development.

## Figures and Tables

**Figure 1 ijms-24-10692-f001:**
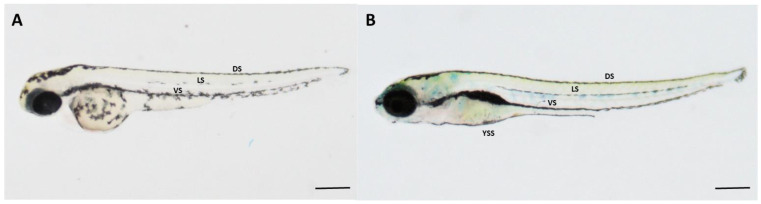
Development of melanocyte pigment cell pattern in zebrafish (*Danio rerio*): (**A**) Embryonic melanocyte pattern, where the first melanocyte strip phenotype could be observed at 2 dpf; (**B**) Early larval melanocyte cell arrangement is complete at 6 dpf and is mainly composed of melanocytes arranged into four stripes: dorsal stripe (DS), lateral stripe (LS), ventral stripe (VS), and the yolk sac stripe (YSS). [Scale bar: 100 µm].

**Figure 2 ijms-24-10692-f002:**
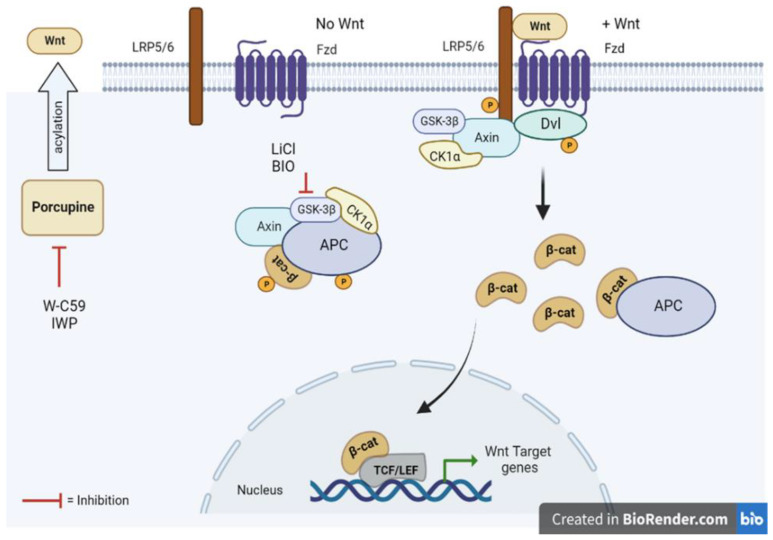
Schematic representation of the canonical Wnt signal transduction pathway. (**Left**) The destruction complex of Axin, APC, GSK-3β, CK1α, and β-catenin is formed when the Wnt ligands are not present in the cytosol, and β-catenin is phosphorylated by CK1α and GSK-3β. (**Right**) In the presence of Wnt ligands, the signalling pathway is stimulated by recruiting Dvl and Axin proteins to the cell membrane and binding to the receptors (Fzd) and co-receptors (LRP5/6) consecutively. This process allows for stabilisation of β-catenin in the cytosol and translocation into the nucleus for signal transduction.

**Figure 3 ijms-24-10692-f003:**
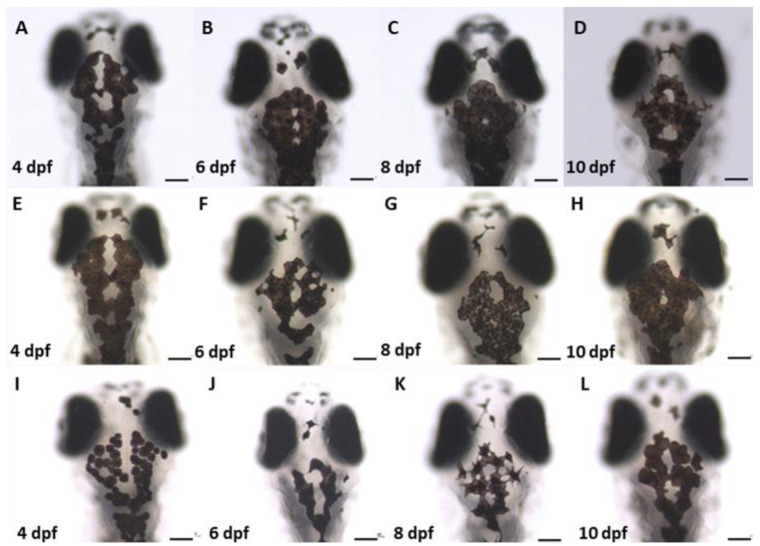
The comparison of melanocyte count in control (**A**–**D**) with LiCl- (**E**–**H**) and W-C59- (**I**–**L**) treated zebrafish larvae at 4 dpf, 6 dpf, 8 dpf, and 10 dpf developmental stages are shown in this figure. It illustrates the differences in melanocyte count in Wnt activator and inhibitor-treated samples compared to the control group at each life stage. [Scale bar: 100 µm].

**Figure 4 ijms-24-10692-f004:**
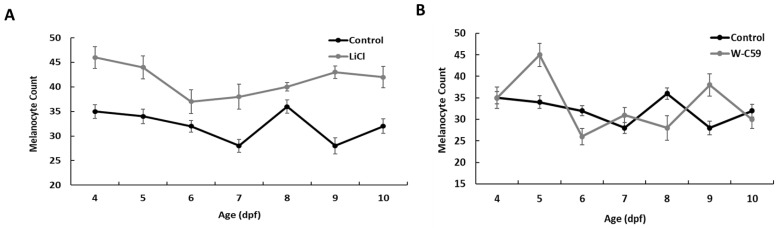
Comparison of the fluctuation in melanocyte count between the control and two treated groups (LiCl and W-C59) from 4 dpf to 10 dpf developmental stages. Fish exposed to LiCl at 10 hpf showed comparatively higher melanocyte counts (**A**), while fish exposed to W-C59 displayed sudden fluctuations in melanocyte densities at each age compared to the un-manipulated control group (**B**).

**Figure 5 ijms-24-10692-f005:**
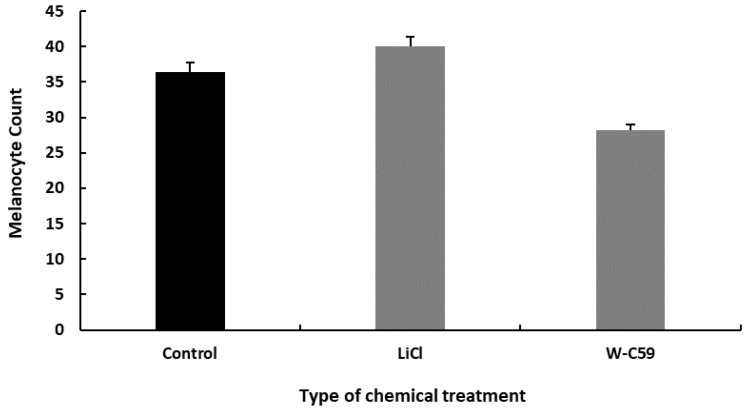
Comparison of the melanocyte count at the 8 dpf developmental stage in control and chemical-treated zebrafish larvae. The bar chart shows the differences in melanocyte development between LiCl and W-C59-treated samples and the control group at the 8 dpf developmental stage.

**Figure 6 ijms-24-10692-f006:**
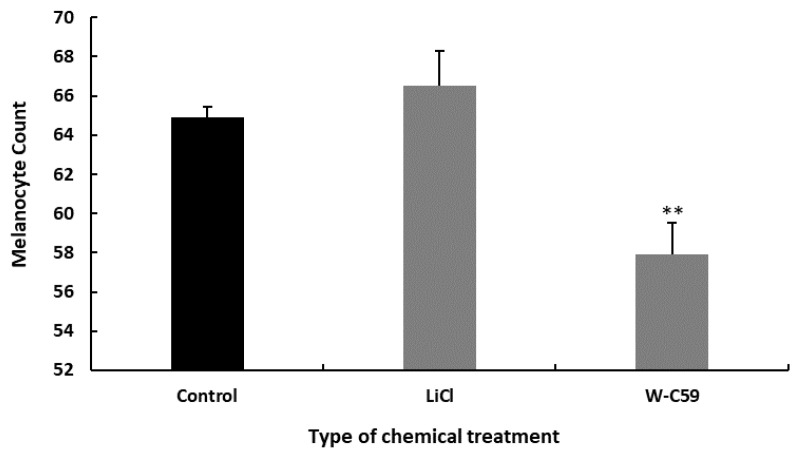
Comparison of the melanocyte count at the mid-larval stage in control and chemical-treated zebrafish larvae. LiCl exposure resulted in an increased melanocyte count, while W-C59-treated fish samples showed a significant reduction in melanocyte count compared to the control group at the mid-larval stage.

**Figure 7 ijms-24-10692-f007:**
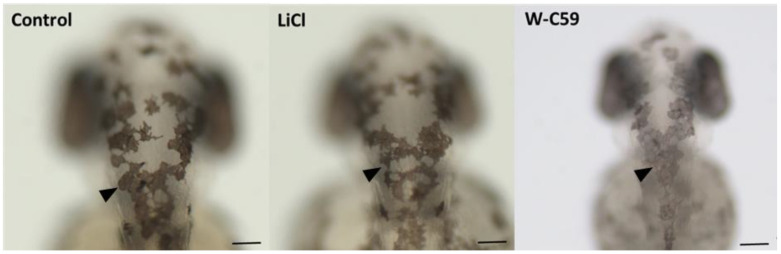
Comparison of the melanocyte phenotypic differences between the control and two treated groups (LiCl and W-C59) at 2 dpf embryonic stage of zebrafish. This figure illustrates the morphological and morphometric differences in melanocytes of untreated control, which were exposed to LiCl and W-C59. [Scale bar: 100 µm].

**Figure 8 ijms-24-10692-f008:**
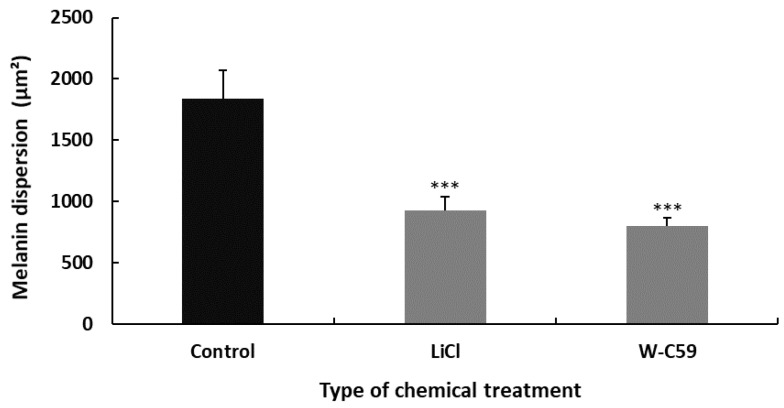
Variations in the melanin dispersion at 2 dpf embryonic stage of control and two treated groups (LiCl and W-C59) of zebrafish. Exposures to LiCl and W-C59 chemicals resulted in significant reductions in melanin dispersion of melanocytes compared to the control group at the embryonic stage.

**Figure 9 ijms-24-10692-f009:**
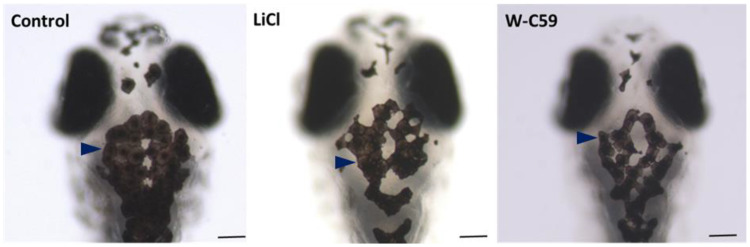
Comparison of the melanocyte phenotypic differences between control and two treated groups (LiCl and W-C59) at 6 dpf embryonic stage of zebrafish [Scale bar: 100 µm].

**Figure 10 ijms-24-10692-f010:**
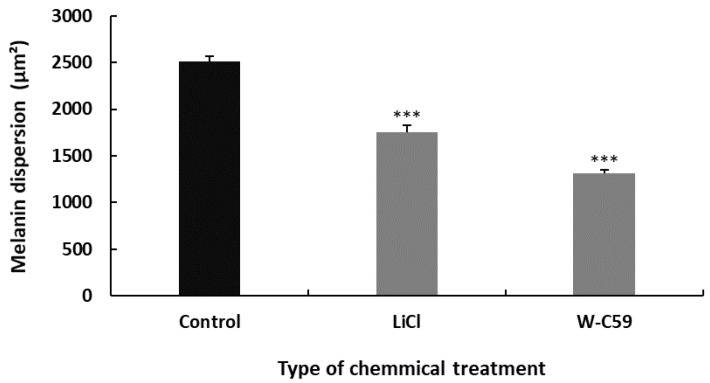
Variations in the melanin dispersion at 6 dpf larval stage of control and two treated groups (LiCl and W-C59) of zebrafish. Exposures to LiCl and W-C59 chemicals resulted in significant reductions in melanin dispersion of melanocytes compared to the control group at the post-embryonic stages.

**Figure 11 ijms-24-10692-f011:**
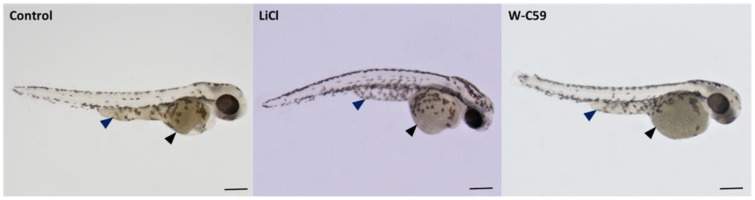
Analysis of migratory defects of melanocytes at 2 dpf embryonic stage of control and two treated groups (LiCl and W-C59) of zebrafish. This figure demonstrates the differences in melanocyte migration of untreated control, and which were exposed to LiCl and W-C59. [Scale bar: 200 µm].

**Figure 12 ijms-24-10692-f012:**
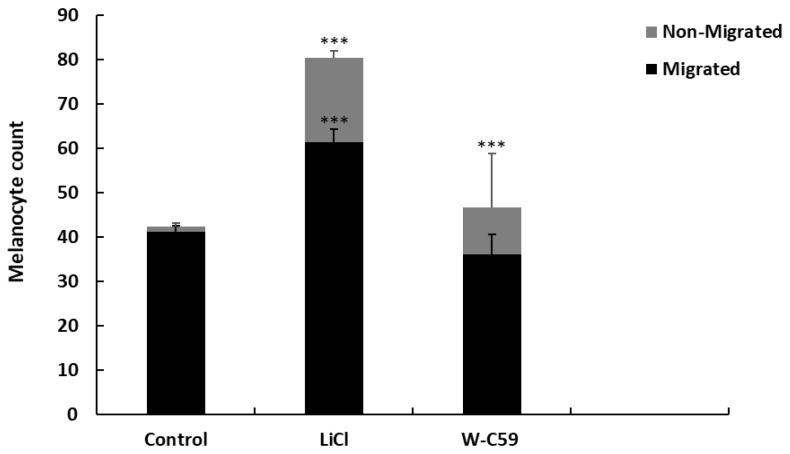
Analysis of migratory defects of melanocytes at 2 dpf embryonic stage of control and two treated groups (LiCl and W-C59) of zebrafish. Chemical exposures resulted in a significant increase in the non-migrated melanocyte count, whereas LiCl exposure showed a significant increase in the migrated melanocyte count and an insignificant reduction in the migrated melanocyte count was exhibited in W-C59 exposed fish embryos.

**Figure 13 ijms-24-10692-f013:**
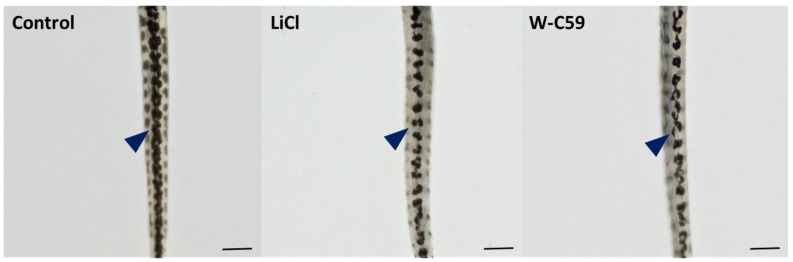
Variations in melanocyte arrangement during dorsal stripe formation at the 6 dpf larval stage of control and two treated groups (LiCl and W-C59) of zebrafish. The figure highlights the changes in the contribution of melanocyte numbers in the dorsal stripe formation of LiCl and W-C59 chemical-exposed larvae compared to the control sample. [Scale bar: 200 µm].

**Figure 14 ijms-24-10692-f014:**
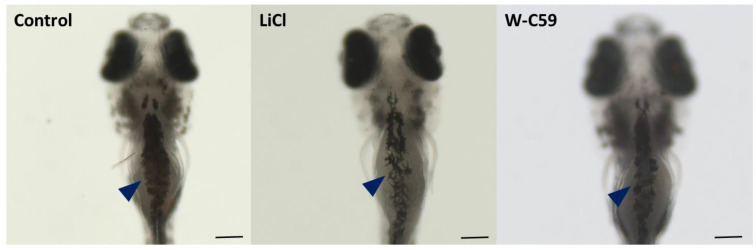
Variations in melanocyte arrangement during yolk sac stripe formation at the 6 dpf larval stage of control and two treated groups (LiCl and W-C59) of zebrafish. The figure highlights the changes in the contribution of melanocyte numbers in the yolk sac stripe formation of LiCl- and W-C59-exposed larvae compared to the control sample. [Scale bar: 200 µm].

**Figure 15 ijms-24-10692-f015:**
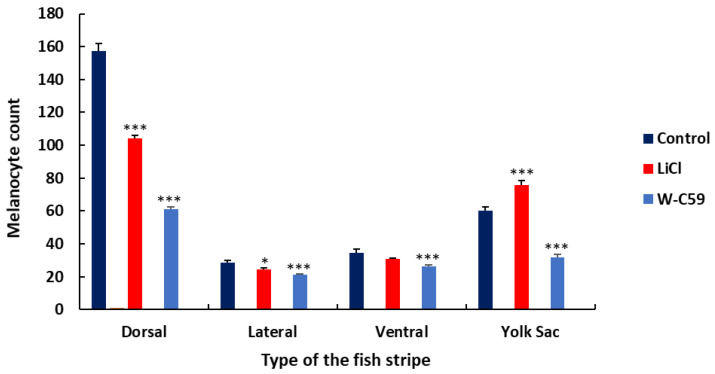
Analysis of melanocyte arrangement defects in zebrafish larvae in response to Wnt chemical modulators. LiCl exposure displayed a significant stimulatory action on melanocyte stripe formation compared to the fish incubated in W-C59, except for ventral stripe formation.

**Figure 16 ijms-24-10692-f016:**
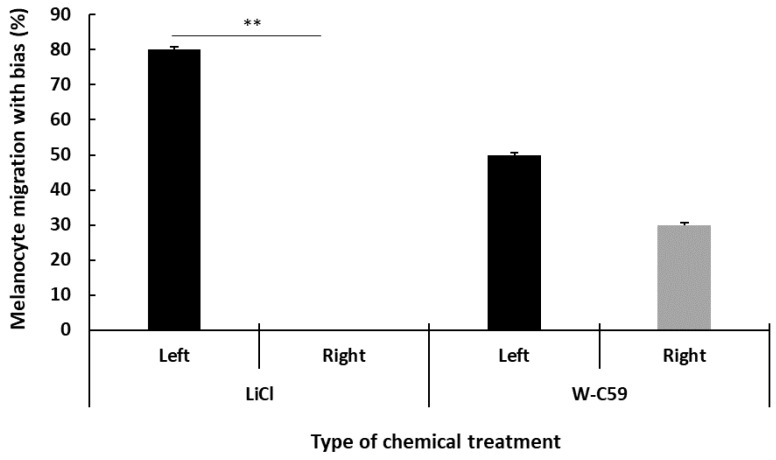
Analysis of biased melanocyte migration along the L–R axis of zebrafish larvae at 6 dpf in response to Wnt chemical modulators. LiCl-exposed fish showed a significant left-right (L-R) asymmetry in melanocyte migration, while no significant L-R bias in melanocyte migration was observed in the W-C59 fish group.

**Figure 17 ijms-24-10692-f017:**
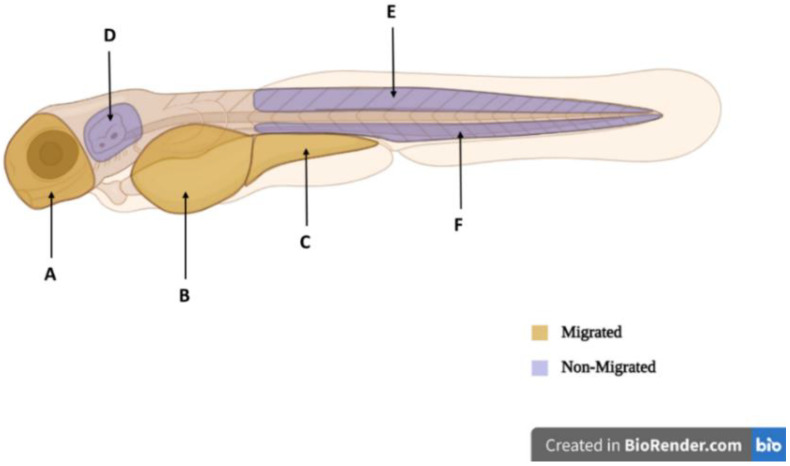
The regions in the embryo used to define migrated and non-migrated melanocytes for quantitative analysis of melanocyte migration are shown in the figure. The yellow areas (**A**–**C**) indicate migrated melanocytes on the anterior head (**A**), yolk sac region (**B**), and yolk sac extension (**C**). The red areas (**D**–**F**) define non-migrated melanocytes near the ear (**D**), in the region between the dorsal stripe and horizontal myoseptum (**E**), and in the region between the ventral stripe and horizontal myoseptum (**F**).

## Data Availability

Data is contained within the article.

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
