# Peer review of "Characterising the Effect of Wnt/β-Catenin Signalling on Melanocyte Development and Patterning: Insights from Zebrafish (Danio rerio)"

_ijms, 2023, doi:10.3390/ijms241310692_

Round 1

Reviewer 1 Report

In this research, Praneeth Silva and Devi Atukorallaya looked into the prenatal effects of wnt signaling pathway on melanophores number, morphology and distribution on embryonic stage zebrafish. They treated zebrafish embryos with wnt enhancer LiCl or inhibitor W-C59 at 10 hpf, then documented melanophores from 2dpf to 15 dpf. Their results confirmed wnt pathway has a positive regulatory effect on melanophore development, prenatally. The concept is new and experiments documented pretty well. I think there is some revision that needs to be done to improve the manuscript.  

Major points:

1.      Please provide evidence of drug efficacy under the given treatment, by RT-PCR, antibody staining or other methods. Since authors stated this is the first time doing so in zebrafish, I assume there’s no reference of such treatment in previous research, so it will be necessary to demonstrate wnt is activated by LiCl and inhibited by W-C59 after treatment.

2.      Please list how many fish were used in each experiment either in figure legends or in method section. Data didn’t show significant differences in Fig2.2, Fig2.3, especially LiCl treated group, but authors still claimed that LiCl had a stimulatory effect on melanogenesis. Given LiCl treated fish had smaller melanophore area, the conclusion is even weaker. One option is to add more replicates and increase N size.

3.      Dispersed pigment makes melanophores difficult to count given their merged cell boundaries and various cell sizes. Please show examples of how you count cell numbers in the case of connected cells like fig2A. Epinephrine is commonly used to contract melanin pigment, especially necessary in this study different treatments seem to have various effects on melanophore morphology which would apparently affect counting results.

4.      How to explain the sudden increase of melanophore density at 8dpf and then fell back at 9dpf? Fig2.2 is redundant to fig2.1 if they are from the same experiment. Please provide the unit for density, otherwise Y should be melanophore count.

5.      L-R asymmetry from LiCl treatment is interesting and worth further investigation and discussion. For example, why left biased? What about organ L-R distribution? Is it recorded in other animal system or zebrafish specific? Is it special to LiCl or the effect of general wnt activation? If not reported, what’s the possible difference and mechanism? I really think the current experiment settings and results are not strong enough for the journal. This observation with some solid follow up experiments and additional making-sense explanations could promote the research.

6.      Please be more careful when making statements without references, or conclusions not covered by the current research. For example:

Line 121, in zebrafish instead of vertebrates, since only zebrafish studied here.

Line 328, “LiCl has not been applied in melanophore research of zebrafish (Danio rerio) as a Wnt signalling pathway enhancer so far.” Not correct. Plenty of studies have used LiCl as wnt activator in zebrafish, and many of them mentioned or even focused on melanophore pigmentation.

Line 348-351, it’s over-interpretated the two drugs were affecting melanophores development, differentiation and survival of stem cells, based on melanophore counts in fig2.2.

Minor points:

Line 36: Wnt1, not wnt3a according to the citation ref7.

Recent progresses of wnt regulating hair pigmentation are not mentioned in the introduction. For example:Nicole R. Infarinato et al., Genes Dev. 2020 Dec 1; 34(23-24): 1713–1734. Qi Sun et al., Nature volume 616, pages774–782 (2023)

Orient images of fish leftwards. Add titles in the figure to indicate different stages or treatments, for example: 2 dpf for fig1A and 6dpf for fig1B, and drug treatments in fig2.

It would make more sense to introduce the mechanisms of the two drugs in the introduction section.

Fig2.4.1 Please describe the actual morphological differences, and with zoom in to demonstrate the pigment size. Note pigment area does not equal to cell. Cell area can be measured by melanophore reporter line, or staining. Pigment dispersion is dynamically changing, which can be affected by many reasons, for example hormone, background adaptation, or nutrition. The other option might be using drug to fully disperse melanin before fixation to measure the effect on cell area. Otherwise, it is better to describe as pigment area.

Fig2.5 Blue and black arrows are hard to distinguish. Figure2.5, number 5 is not bold.

Put figure 1.2 next to figure2.5 for the convenience of comparison. Since non-migrated sites have even higher melanophore counts maybe it’s because overall increased number around yolk instead of migratory effect? Please provide reference for the definition of migratory sites vs. non-migratory sites.

Fig2.7.1 images are too dark to see the morphology.

Fig2.7.2 dorsal/lateral stripe melanophore counts are significantly lower in LiCl treated group, which seems contradictory to the conclusion.

Fig2.8 Please provide example images of L-R asymmetry of LiCl treated fish.  

Wnt signaling functioning on melanophore has been extensively studied in various animals including zebrafish. The major contribution of this study is the effect of prenatal treatment, which is not discussed enough in the Discussion section.

Overall the language is good. There's some redundancy. Some paragraphs could be combined. And please be more cautious with terminologies. 

Author Response

Major points:

  1. Please provide evidence of drug efficacy under the given treatment, by RT-PCR, antibody staining or other methods. Since authors stated this is the first time doing so in zebrafish, I assume there’s no reference of such treatment in previous research, so it will be necessary to demonstrate wnt is activated by LiCl and inhibited by W-C59 after treatment.

Thank you very much for taking the time to revise the manuscript. In this research article, we have attempted to highlight the phenotypic effect of the Wnt cell signaling pathway on melanocyte development. We have already performed whole-mount in situ hybridization for molecular detection of Wnt expression. However, due to the abundance of results, we have planned to publish them in a separate article.

  1. (a) Please list how many fish were used in each experiment either in figure legends or in method section.

This was corrected in the Lines #509 -510

(b) Data didn’t show significant differences in Fig2.2, Fig2.3, especially LiCl treated group, but authors still claimed that LiCl had a stimulatory effect on melanogenesis. Given LiCl treated fish had smaller melanophore area, the conclusion is even weaker. One option is to add more replicates and increase N size.

Thank you very much for this comment. According to the comment following changes were made in the manuscript.

In the Lines #350

“To our knowledge, this study is the first to demonstrate that the early embryonic exposure of zebrafish (Danio rerio) to Wnt chemical modulators, including LiCl and W-C59, during the early neurulation period, has an impact on the generation of melanocytes”

According to the reviewer's comments, we have removed the word "significant" from the results section under Figure 2.2 and Figure 2.3, as there was an increase in melanocyte density in LiCl-treated fish compared to the control and W-C59 groups. Both Figure 2.2 and 2.3 demonstrate a stimulatory effect on melanocyte development after prenatal exposure to LiCl. This has been further described in the discussion section, specifically in lines #361 and from #371-#373.

  1. Dispersed pigment makes melanophores difficult to count given their merged cell boundaries and various cell sizes. Please show examples of how you count cell numbers in the case of connected cells like fig2A. Epinephrine is commonly used to contract melanin pigment, especially necessary in this study different treatments seem to have various effects on melanophore morphology which would apparently affect counting results.

According to the literature, there are several techniques for counting cell numbers, such as using epinephrine chemicals and conducting microscopic software analysis. In our study, we employed the second method mentioned in the methodology section 4.3. We utilized ZEN 2011 software (blue edition, Zeiss, Germany, 2011) for counting melanocytes. By utilizing software tools, melanocytes within the region of interest could be automatically detected.

Methodology section was revised in the Line #519

  1. How to explain the sudden increase of melanophore density at 8dpf and then fell back at 9dpf? Fig2.2 is redundant to fig2.1 if they are from the same experiment. Please provide the unit for density, otherwise Y should be melanophore count.

The reason for the sudden increase of melanophore density at 8th day of fish development has been descriptively explained from the lines #354-#358 by revising the paragraph.

The transition from one life stage to another results in the degeneration of existing melanocytes and the addition of new melanocytes to the fish. According to previous literature on fish melanocyte development, the decline in melanocyte number at 9 dpf could be attributed to the loss of melanocytes in higher numbers than the newly produced melanocytes.

Both Figure 2.1 and Figure 2.2 are important as they highlight crucial points. In Figure 2.1, we aimed to compare the fluctuation in melanocyte count between the control and two treated groups (LiCl and W-C59) from 4 dpf to 10 dpf developmental stages. In Figure 2.2, we specifically focused on melanocyte formation at the 8 dpf developmental stage, which is considered a critical stage in postembryonic growth and melanocyte development based on previous studies and the results of the current study.

The titles of the y-axes in Figure 2.1, 2.2, and 2.3 have been renamed to "Melanocyte Count."

  1. L-R asymmetry from LiCl treatment is interesting and worth further investigation and discussion. For example, why left biased? What about organ L-R distribution? Is it recorded in other animal system or zebrafish specific? Is it special to LiCl or the effect of general wnt activation? If not reported, what’s the possible difference and mechanism? I really think the current experiment settings and results are not strong enough for the journal. This observation with some solid follow up experiments and additional making-sense explanations could promote the research.

The entire discussion section has been revised in the newly revised manuscript, addressing each comment highlighted here.

From Line #450-453 and #461-478

  1. Please be more careful when making statements without references, or conclusions not covered by the current research. For example:

(a) Line 121, in zebrafish instead of vertebrates, since only zebrafish studied here.

Corrected according to the comment.

(b) Line 328, “LiCl has not been applied in melanophore research of zebrafish (Danio rerio) as a Wnt signalling pathway enhancer so far.” Not correct. Plenty of studies have used LiCl as wnt activator in zebrafish, and many of them mentioned or even focused on melanophore pigmentation.

This statement has been removed in the revised manuscript.

(c) Line 348-351, it’s over-interpretated the two drugs were affecting melanophores development, differentiation, and survival of stem cells, based on melanophore counts in fig2.2.

This interpretation was done based on the analysis of previous literature, as changes in melanocyte number are always related to changes in melanocyte stem cell number.

 Minor points:

(d) Recent progresses of wnt regulating hair pigmentation are not mentioned in the introduction. For example:Nicole R. Infarinato et al., Genes Dev. 2020 Dec 1; 34(23-24): 1713–1734. Qi Sun et al., Nature volume 616, pages774–782 (2023)

This was not addressed in the introduction section as hair pigmentation is not directly associated with our current study.

(e) It would make more sense to introduce the mechanisms of the two drugs in the introduction section.

According to the reviewer’s suggestion, mechanisms of LiCl and W-C59 drugs were inserted in the introduction section from Line #104-111

Fig2.4.1 Please describe the actual morphological differences, and with zoom in to demonstrate the pigment size. Note pigment area does not equal to cell. Cell area can be measured by melanophore reporter line, or staining. Pigment dispersion is dynamically changing, which can be affected by many reasons, for example hormone, background adaptation, or nutrition. The other option might be using drug to fully disperse melanin before fixation to measure the effect on cell area. Otherwise, it is better to describe as pigment area.

The term "cell area" has been replaced with "pigment dispersion" in the revised manuscript

Fig2.5 Figure2.5, number 5 is not bold.

Corrected in the revised manuscript

Put figure 1.2 next to figure2.5 for the convenience of comparison. Since non-migrated sites have even higher melanophore counts maybe it’s because overall increased number around yolk instead of migratory effect? Please provide reference for the definition of migratory sites vs. non-migratory sites.

Inserted the reference in the Lines #539

Fig2.7.1 images are too dark to see the morphology.

These images were captured in bright field mode under the stereo microscope. Here, our objective was to demonstrate the overall variations of the yolk sac stripe in response to chemical treatments, rather than focusing on individual melanocyte morphology within the yolk sac stripe.

Fig2.7.2 dorsal/lateral stripe melanophore counts are significantly lower in LiCl treated group, which seems contradictory to the conclusion.

This phenomenon has been discussed in the discussion part.

Wnt signaling functioning on melanophore has been extensively studied in various animals including zebrafish. The major contribution of this study is the effect of prenatal treatment, which is not discussed enough in the Discussion section.

This has been corrected in the discussion section from Line #328-#332 and #347-#352

Reviewer 2 Report

"This is an interesting study on the role of Wnt signaling in melanocyte development and differentiation. Since Wnt signaling has been extensively researched in zebrafish, including melanocytogenesis, a bit more credit should be given to others in the introduction. The use of LiCl in this process is novel and of particular note is the observation of disrupted sidedness of melanocyte migration patterns when the pathway is modulated by chemichal exposure! This is a really interesting observation! I have no doubt that this article will be of interest to the IJMS readership, as it is well organized, well written, and the tests and their interpretation are sound. Congratulations on conducting a nice study! There are a few minor points I would like to suggest to improve the overall quality for publication in IJMS.

Specific Comments:

Introduction, line 30: The genome contains many more than 25 Wnt genes! I think you meant 'contains' instead of 'consists.' ;-)

Line 90 (and elsewhere in manuscript): I caution against the use of 'prenatal.' I understand that you mean to use zebrafish as a model for human development. However, since you are studying embryonic and larval development in fish, it would be more appropriate to refer to these stages as being tested rather than 'prenatal.' Comparing zebrafish embryonic / larval stages to human prenatal development would be more appropriate in the discussion.

Overall, you should acknowledge more and give credit in the introduction to others who have already explored the role of Wnt signaling in melanocyte development before you. You can point out (discussion?) that while these studies focused on specific Wnts, your approach likely affected the pathway in a broader sense.

Figure 1: By convention, anterior should be on the left. Can you please flip these panels? Also, the resolution of these images is low and blurry. Can you retake these pictures? You could also try reducing the white space and increasing larval views, or add insets with higher magnifications to show details.

Figure 1.1: Can you add LiCl and its stimulatory effect to this illustration? W-C59 is already included; it would help readers understand the molecular interactions of (ant)agonists.

Figure 2.0: The background (ventral melanocytes) obscures the dorsal melanocytes. Would a different orientation (e.g., side view) not convey the point better? Can the image quality be improved, especially for panels A - D? A similar problem is observed in Figure 2.7.1. Reducing the white space could help balance the exposure/dark/light distribution better. Please consider retaking the images.

Figure 2.4.2: What do the arrowheads indicate? Please rephrase: The comparison shows representative (hopefully ;-) ) specimens taken from groups, you are not comparing groups here. The figure legend is cryptic and repeats the same statement twice. The comparison of.... illustrates differences. One of the two statements is redundant and should rather specifically spell out the difference (i.e., result) being shown! This holds true, as far as I can tell, for all figure legends. Please rephrase them and state the specific result you are illustrating.

Section 2.5: Morphology/Morphometry. Is there any data about melanocyte dendrites, especially when comparing migration patterns? Figure 2.5: The images are blurry and of low resolution. You can improve them further by reducing the white space and adding insets for higher detail. What are the arrowheads pointing at?

Figure 2.6: Again, the legend is cryptic. Are these columns stacked or grouped?

Fig. 2.7: Reduce the white space. If the arrowhead is supposed to point out the midline, it should probably be oriented up or down, not sideways.

Fig. 2.7.1: As mentioned before, there is poor contrast. Can you please improve it?

Line 328: I think it would be appropriate to cite your FASEB paper here (LiCl has not been used before, except by you).

The discussion should be a little more concise as it sounds a bit repetitive. In the section between lines 382-392, I suggest you also discuss (referring back to the introduction) how you modulate Wnt more broadly by LiCl or W-C59 exposure, compared to studies that focused on specific Wnt functions (e.g., Wnt3a) through overexpression or mutations. You should also discuss whether or not you observed secondary Wnt targets unrelated to melanocytes due to the chemichal exposure approach. Lastly, there have been studies regarding the sidedness of the zebrafish CNS. You could discuss how Wnt correlates or potentially interacts with these other pathways (e.g., Nodal) at this late stage.

Figure legends: All figure legends should be revised. Ideally, the title of the legend should state the actual result being illustrated. The body of the figure legends should explain what the reader is supposed to see without them having to read the main text again. Please describe the arrowheads and other labels!

Reviewer 3 Report

This manuscript describes the effect of Wnt signalling  activity on zebrafish melanocyte development and patterning. The results showed and established an important role of Wnt signalling in melanocyte lineage and emphasises the importance of a balanced Wnt signalling level for proper melanocyte development and patterning. The work is interesting, carefully performed, and provide new and significant information to researchers working on developmental biology. Statistical analyses was adequately performed and there are no doubts about the results.

Comments

1) Please renumber figure 1.2. as this figure appeared at the end of the manuscript.

2) The effect of LiCl appears significant. The authors found that in LiCl-treated fish, the highest melanocyte accumulations were recognised in both migrated sites. This finding needs some more discussion and explanation.

3) The importance of Wnt signalling in the development of melanocytes in zebrafish is highlighted. The authors can compare their finding with other model organisms or human cell lines.

Author Response

Thank you very much for your comments.

Comments

  • Please renumber figure 1.2. as this figure appeared at the end of the manuscript.

Corrected as Figure 3.0

  • The effect of LiCl appears significant. The authors found that in LiCl-treated fish, the highest melanocyte accumulations were recognised in both migrated sites. This finding needs some more discussion and explanation.

This has been already explained in the discussion part

3) The importance of Wnt signalling in the development of melanocytes in zebrafish is highlighted. The authors can compare their finding with other model organisms or human cell lines.

This is the first study conducted thus far on the effects of prenatal exposure to Wnt regulatory chemicals on melanocyte formation, phenotypic development, migration, and patterning.

Reviewer 4 Report

Dear Authors,

In my opinion, the manuscript entitled “Characterizing the Effect of Wnt/β-catenin Signalling on Melanocyte Development and Patterning: Insights from Zebrafish (Danio rerio)” is acceptable for publishing in IJMS.

The study was designed to investigate the effect of Wnt signaling activity on zebrafish melanocyte development and patterning. The work is interesting, well design and well presented.

Comments:

Kindly revise reference format according to the IJMS author guideline.

Author Response

Comments:

Kindly revise reference format according to the IJMS author guideline.

Thank you very much for your comments.

In our article, all the references have been numbered in order of appearance in the text (including table captions and figure legends) and listed individually at the end of the manuscript. According to the IJMS referencing guidelines, all the references have been prepared with the bibliography software package of EndNote.

Round 2

Reviewer 1 Report

Though some questions not fully answered, it's still an interesting research to a broad readership, with solid experiments, observations and quantifications.  I recommend to accept the manuscript.